



# 1 What is climate change doing in Himalaya? Thirty
# 2 years of the Pyramid Meteorological Network (Nepal)

Franco Salerno[1,2,*,†], Nicolas Guyennon[3,**,†], Nicola Colombo[4], Maria Teresa Melis[5,6],
Francesco Gabriele Dessi[5,], Gianpietro Verza[5], Kaji Bista[7], Ahmad Sheharyar[1], Gianni
Tartari[2]
1 National Research Council, Institute of Polar Sciences, ISP-CNR, Milan, Italy
2 National Research Council, Water Research Institute, IRSA-CNR, Brugherio (MB), Italy;
3 National Research Council, Water Research Institute, IRSA-CNR, Montelibretti (Roma), Italy;
4 Department of Agricultural, Forest and Food Sciences, University of Turin, Grugliasco, Italy
5 Ev-K2-CNR, Bergamo, Italy;
6 Departmento of Chemical and Geological Sciences, Univerity of Cagliari, Monserrato (CA), Italy
7 Nepal Academy of Science and Technology (NAST), Kathmandu, Nepal
* Correspondence: franco.salerno@cnr.it; **nicolas.guyennon@irsa.cnr.it
† Franco Salerno and Nicolas Guyennon equally contributed to this paper.

## 18 Abstract

Climate change is deeply impacting mountain areas around the globe, especially in
Himalaya. However, the lack of long-term meteorological observations at high elevations
poses significant challenges to understand and predict impacts at various scales. This also
represents a serious limit for model-based projections of future behavior of crucial
elements of the mountain cryosphere such as glaciers. Here, we present the Pyramid
Meteorological Network, located in Himalaya (Nepal), on the southern slopes of Mt.
Everest. The network is composed of 7 meteorological stations located between 2660 and
7986 m a.s.l., which have collected continuous climatic data during the last 30 years
(1994-2023). In this paper, details are provided regarding instrument types and
characteristics as well as data quality control and assessment. The obtained data series are
available on a newly created geoportal. We leverage these unique records to present new
knowledge on the Himalayan climate, benefiting also from the highest observational
climatic series in the world (Pyramid station, located at above 5000 m a.s.l., close to
Khumbu Glacier). These data will provide fundamental knowledge on climate dynamics
in Himalaya that will inform research at high elevations in the coming years. The dataset



is available freely accessible from https://geoportal.mountaingenius.org/portal/
(https://zenodo.org/records/14450214) (Salerno et al., 2024).
1 **Introduction**
Global temperature has been increasing at unprecedented rates during the Anthropo-
cene, impacting both natural and human systems (e.g., Mukherji et al., 2023). Alpine bi-
omes, among the most sensitive natural ecosystems to climate warming, show rapid shifts
of species distribution ranges and modulations of species interactions (e.g., Sigdel et al.,
2021). Himalayan glaciers have been losing mass in the last decades (Biemans et al.,
2019). The current uncertainties concerning the glacial shrinkage in the Himalayas are
mainly attributed to the lack of measurements of climatic forcings (e.g., Bhattacharya at
al., 2021). Indeed, recent research has underlined the need for fine scale investigations,
especially at high elevation, to better model the glacio-hydrological dynamics (Yao et al.,
2022). In addition, according to Yang et al. (2018), reliable meteorological data at glacial
elevations are essential to: (1) place the observed glacial changes in the context of current
climatic change, (2) understand hydro-meteorological relationships in cryospheric envi-
ronments, and (3) calibrate dynamically and statistically downscaled climate fields. How-
ever, there are few high-elevation weather stations where the glaciers are located, espe-
cially in Himalaya. This can be attributed to the remote location of glaciers and the rugged
terrain, which make physical access difficult (e.g., Salerno et al., 2015; Lin et al., 2021).
As a consequence of the remoteness and difficulty in accessing several high-elevation
sites combined with the complications of operating automated weather stations (AWSs)
in remote areas, long-term measurements are challenging (Yang et al., 2018). For in-
stance, in Himalaya, meteorological stations at high elevations are extremely scarce
(Mountain Research Initiative EDW Working Group, 2015; Salerno et al., 2015; T. Mat-
thews et al., 2020). Therefore, in several studies, climatic data at high elevations had to
be estimated using low-elevation data (Shrestha et al., 2014; Zhang et al., 2015), which
are more common. This is the case of the central Himalaya, where the Department of
Hydrology and Meteorology of Nepal (www.dhm.gov.np/) maintains more than 300 long-
term rain stations, although they are mainly located below 3000 m a.s.l..
In this context, in the early 1990s, the Pyramid Meteorological Network was created
by the Italian *Ev-K2-CNR Committee* (www.evk2cnr.org). This network is composed of



7 automatic weather stations located on the southern side of Mt Everest (along the
Khumbu Valley), in the central Himalaya (Sagarmatha National Park - SNP; Amatya et
al., 2010; Salerno et al., 2010) ranging from 2660 to 7986 m a.s.l.. For each station, the
following variables are collected on an hourly basis: air temperature, total precipitation,
relative humidity, atmospheric pressure, and wind speed and direction.
Here, we present the database in which all meteorological data are stored, freely ac-
cessible from https://geoportal.mountaingenius.org/portal/ (https://zenodo.org/rec-
ords/14450214), and we explore the small-scale climate variability of the longest time
series of the network, the Pyramid station (5035 m a.s.l.), located close to the Khumbu
Glacier.

## 76   2 Region of investigation

Salerno et al., 2015 describes the ground network of automatic weather stations
(AWSs) belongs to the Pyramid Meteorological Network, which is located on the
southern side of Mt Everest (along the Khumbu Valley), in central Himalaya (Sagarmatha
National Park - SNP; Amatya et al., 2010; Salerno et al., 2010) (Fig. 1). The land-cover
classification shows that almost one-third of the territory is characterised by glaciers and
ice cover, while less than 10% of the park area is forested (*Abies spectabilis*, *Betula utilis*)
(Magnani et al., 2018; Pandey et al., 2020). The tree line is located at approx. 4050 m
a.s.l., while the landscape is dominated by alpine tundra and lichen above this elevation
(Bhuju et al., 2010; Sigdel et al., 2021). Glacial surfaces are distributed from 4300 to
above 8000 m a.s.l. Around 75% of the glacier surfaces are located between 5000 and
6500 m a.s.l. (Thakuri et al., 2014, 2016), and ca. 25% of the glacierised area is debris-
covered (Shea et al., 2015; Salerno et al., 2017). Glaciers in this area are classified as the
summer-accumulation type, which are fed mainly by summer monsoon precipitation
(Tartari et al., 2008).
The climate in the South Asia and Himalayan region has a strong annual cycle, with
the South Asian monsoon that is a phase of this annual cycle. During the pre-monsoon
season (MAM), the westerlies prevail over this region and are deflected when crossing
the Himalayan mountains. During the monsoon season (JJAS), the westerlies move
northward, while south-westerly flows dominate the upper level and southeasterly flows
from Bay of Bengal dominates the lower level (Ichiyanagi et al., 2007). After the offset



of the monsoon, the south-westerly and southeasterly flows are replaced by the westerlies.
The warm area moves to the south and both air temperature and humidity decrease
considerably. Cooling and drying are further enhanced towards the winter (Yang et al.,
100 2018).

Regarding the precipitation, the measurements at Pyramid station (Z5035) show that
90% is concentrated from June to September, while the probability of snowfall during
these months is very low (4%); the annual cumulated precipitation at this elevation is 446
mm, with a mean annual temperature of –2.5 °C (Salerno et al., 2015). Precipitation
linearly increases to an elevation of 2500 m a.s.l. and exponentially decreases at higher
elevations (Salerno et al., 2015). Finally, the wind regime of the area is characterised by
up-valley winds during the day throughout the year, while weak up-valley winds occur at
night during the monsoon season, with some evidence of down-valley winds occurring at
night in the winter (Potter et al., 2018 and references therein reported). Strong diurnal
katabatic winds also occur at the higher elevations (above ca. 4500 m a.s.l.) due to
enhanced glacier melting under warm atmospheric conditions (Salerno, et al., 2023).
**3 Data and methods**
**3.1 Weather stations**
The first automatic weather station (AWS0) was installed in October 1993, near the
Pyramid Laboratory, at 5035 m a.s.l. (Fig. 1, 2; Bertolani et al., 2000). AWS0 recorded
temperature data until December 2005. A new station (AWS1) was installed just a few
tens of meters away from AWS0 and it has been operating since October 2000. The other
stations were installed in the following years in the Khumbu Valley (Fig. 1, Tab. 1). In
2008, the network included seven monitoring sites, including the highest weather station
of the world, located at South Col of Mt. Everest (7986 m a.s.l.). The locations of all
stations are presented in Figure 1, while Figure 3 shows the temporal availability of the
meteorological data. AWS3 (Z2660) and AWS5 (Z3570) are located below the tree line,
while AWS2 (Z4260) is located close to the upper limit of the vegetation. At higher ele-
vation, AWS0 and AWS1 (Z5035) are close to the glacier front elevation, AWS4 (Z5600)
is situated at the mean elevation of glaciers, and AWSCC (Z7986) characterises the high-
est peaks. The list of measured variables, sensors, manufacturer and accuracy for each
station is presented in Table 3.



Recently, a new meteorological network was established in the Khumbu Valley by the
2019 National Geographic and Rolex Everest Expedition
(https://datadash.appstate.edu/high-altitude-climate/#download), with 5 stations ranging
from 3810 to 8430 m a.s.l. (Matthews et al., 2020). On average, this network is located at
an elevation higher than the Pyramid Meteorological Network, representing mainly the
accumulation zone of the glaciers in the region. Moreover, the GLACIOCLIM group
manages some stations on Changri Nup and Mera Glacier (Wagnon et al., 2021).

**3.2 Geoportal structure**
Since the early 1990s, when the Pyramid Meteorological Network was created, the Ev-
K2-CNR Committee has promoted the sharing of data collected from high-elevation
AWSs. In 2014, the first data sharing system was born, and it was called SHARE (Station
at High Altitude for Research on the Environment) Geonetwork. The system collected
data from 15 stations spread across four countries (Nepal, Pakistan, Italy, and Uganda).
The system was designed for open data management, in line with international directives
and standards for free access to environmental data. Furthermore, based on a
customisation of the GeoNetwork software system, a hierarchical database of the
individual stations and sensors was created (Melis et al., 2013; Locci et al., 2014).
In the last ten years, this web platform has been improved according to new digital
standards and software release. Furthermore, the publication of station data was
accompanied by a new web-GIS platform to provide three services: 1) a structured
metadata and data archive, 2) a simplified interface to provide access to AWSs' data, and
3) a dedicated webGIS platform for geo-referenced data. The new GeoPortal is accessible
at the address https://geoportal.mountaingenius.org/portal/.
An exclusive function provided by the GeoPortal is the direct access to dataset and
databases through a dedicated search data-tab in the portal main menu. Dataset acquired
by the projects are stored in a PostgreSQL DBMS: registered GeoPortal users can query
the data by the Search Data command, and in the results page, it is possible to proceed to
direct download dataset in csv format or to directly consult them in forms of tables and
charts. All information provided by the portal is supplied with their relative metadata. The
metadata database is the core of system: only through metadata it is possible to search
and retrieve resources. The main search window allows to search any string occurrence





in the metadata database: through the result page it is possible to access directly to the
metadata sheet with description of resource. Here it is possible to retrieve the direct
connection with dataset with the possibility of a direct download of the supplied dataset,
accordingly with the file format. Metadata and datasets are strictly related with a two-
ways connection.

**3.3 Data gap filling for temperature and precipitation time series**
Pyramid station has suffered a percentage of missing daily values of ca. 10% and
15% for temperature and precipitation, respectively (Table 2). In this study, we applied
the same gap filling method (quantile mapping) used for missing data in Salerno et al.
(2015), but extending the time series to 2023. All the stations belonging to the network
were tested and used for filling the gaps according to a priority criterion based on the
degree of correlation among data. AWS1 was chosen as the reference station given the
length of the time series and the fact that it is currently still operating. The selected filling
method is a simple regression analysis based on quantile mapping (e.g., Déqué, 2007;
Themeßl et al., 2012). This regression method has been preferred to more complex
techniques, such as the fuzzy rule-based approach (Abebe et al., 2000) or the artificial
neural networks (Abudu et al., 2010; Coulibaly and Evora, 2007), considering the
peculiarity of this case study where all stations are located in the same valley (Khumbu
Valley). This aspect confines the variance among the stations to the elevational gradient
of the considered variable, which can be easily reproduced by the stochastic link created
by the quantile mapping method. In case all stations registered a simultaneous gap, we
applied a multiple imputation technique (Schneider, 2001) that uses some other proxy
variables to fill the remaining missing data. The uncertainty introduced by the filling
process on the Sen's slope (SS) was estimated through a Monte Carlo uncertainty
analysis. Details on the reconstruction procedure and the computation of the associated
uncertainty are provided in Salerno et al. (2015).
**3.4 Statistical analysis**
In this study, the Mann–Kendall test (MK, Kendall, 1975) was applied at the
monthly scale (after daily data aggregation) to analyse the non-stationarity of meteoro-
logical data. This test is widely adopted to assess significant trends in



hydrometeorological time series (Guyennon et al., 2013). This test is non-parametric, thus
being less sensitive to extreme sample values and is independent from the hypothesis
about the nature of the trend, whether linear or not. The MK test verifies the assumption
of the stationarity of the investigated series by ensuring that the associated normalized
Kendall's tau-b coefficient, $\mu(\tau)$, is included within the confidence interval for a given
significance level (for $\alpha = 5\%$, the $\mu(\tau)$ is below $-1.96$ and above $1.96$). We used the Sen's
slope (SS) proposed by Sen (1968) as a robust linear regression allowing the quantifica-
tion of the potential trends revealed by the MK. The significance level is established for
$P < 0.05$. We defined a slight significance for $P < 0.10$. The uncertainty associated with
the SS (1994–2013) is estimated through a Monte Carlo uncertainty analysis (James and
Oldenburg, 1997). In the sequential form (seqMK) $\mu(\tau)$ the test is applied forward starting
from the oldest values (progressive trend) and backward starting from the most recent
values (retrograde trend). The crossing period allows us to identify the approximate start-
ing point of the trend. In this study, the seqMK is applied to monthly vectors. Monitoring
the seasonal non-stationarity, the monthly progressive $\mu(\tau)$ is reported with a pseudo
color code, where the warm colors represent the positive slopes and cold colors the neg-
ative ones

## 4 Results and discussion

At 5035 m a.s.l., the precipitation is concentrated during June–September (around
90%, Fig. 4) and, considering that the mean daily temperature during these months is
above $+0$ °C, we can infer that during these months the probability of snowfall is very
low. According to Salerno et al. (2015), the underestimation of precipitation fallen as
snow during the other months should not be over 20%. Sustained by this analysis, the
trend analysis of precipitation was focused to the warmest months (Fig. 5d).
Trend analysis at high elevation
Figure 5 shows the reconstructed Pyramid time series for Tmin, Tmax, Tmean, and
Prec, after the gap filling procedure. These daily time series for the 1994-2023 period are
available at https://geoportal.mountaingenius.org/portal/. These data, until 2020, have
been presented in Salerno et al. (2023). In this paper, the last three years have been added
to the time series and now we present the results of the last 30 years (1994-2023).

*Maximum air temperature (Tmax)*

During the warm season (from May to October), Tmax shows a significant negative trend (-0.31 ± 0.015 °C y$^{-1}$, p < 0.05) as highlighted by the progressive μ(τ) trend in the bottom graph (full line in orange, Fig. 5a). Increases (although not significant) are observed in November and December; generally, the cold season (from November to April) shows no trend (-0.006 ± 0.013 °C y$^{-1}$, p > 0.1) (full line in blue). On the annual scale the trend is negative, but not significant (-0.022 ± 0.011 °C y$^{-1}$, p > 0.1). The decreasing trend seems to have started in 2007 for the warm season, while in the previous years the negative trend was restricted to only MJJ months, whereas the cold months shows a later start of the decreasing trend, i.e. from 2011.

*Minimum air temperature (Tmin)*

November (+0.06 °C y$^{-1}$, p < 0.05) and December (+0.08 °C y$^{-1}$, p < 0.01) present the highest increasing trend, i.e., both months experienced ca. +2.1 °C over thirty years (Fig. 5c). The cold season experienced a positive trend (0.046± 0.012 °C y$^{-1}$, p < 0.01) mainly concentrated in the post-monsoon period. As highlighted by the progressive μ(τ) trend in the bottom graph (full line in blue), this trend started increasing in 2005. In the warm season, the trend is much lower (0.024± 0.014 °C y$^{-1}$, p < 0.1) and it is negative in May. On the annual scale the trend is moderately negative (-0.030 ± 0.009 °C y$^{-1}$, p > 0.01).

*Mean air temperature (Tmean)*

Figure 5b presents, as expected, intermediate conditions for Tmean. The cold season shows increasing trends, although not significant (0.020± 0.009 °C y$^{-1}$, p > 0.1), and only November and December significantly rise. In the warm season, there is no trend (0.003± 0.010 °C y$^{-1}$, p > 0.1), while the temperature in May decreases. Also considering the annual scale, there is no trend in the last 30 years (0.005± 0.007 °C y$^{-1}$, p > 0.1).

*Total precipitation (Prec)*

In the last years, for all months of the warm season, an overall and strongly significant decreasing trend of Prec has occurred (Fig. 5d). Considering all period, a continuous decreasing trend has occurred since 2000, which became significant at the beginning of 2005. The decreasing Prec trend is highest in August. During the warm season, the reduction of precipitation has been 41%.




## Conclusion

Glaciers in the Himalaya are a major focus of international research, for their relevance
for water and people, their role in climate-land feedbacks and their iconic and not entirely
understood patterns of changes. One of the major drawbacks in Himalayan research of
the cryosphere is that there are almost no long-term climate measurements at high
elevations, where glaciers are located. Here, we presented station data of 7 stations
belonging to the Pyramid Meteorological Network managed by EV-K2-CNR.
Moreover, we presented the precipitation and temperature time series based on a three-
decade effort to ensure a continuous monitoring of the high-elevation climate in the
Himalaya.
Strikingly, our measurements reveal a local cooling at glacierized elevations which is
in stark contrast to the postulated temperature increases. An interpretation of this
phenomenon was provided recently by Salerno et al., 2023. What is interesting here is to
highlight that by means of this unique data and the perseverance of the measurements it
has made it possible to tell a story that goes against the trend of current knowledge based
on data collected elsewhere or at low altitude.
We are convinced that making this data available will open new perspectives on
climate change and its effects in the Himalaya that will guide research at high elevations
in the coming decades

## Data availability

All datasets described and presented in this paper can be openly accessed from
https://geoportal.mountaingenius.org/portal/. Moreover, the dataset is accessed from
https://zenodo.org/records/14450214 (Salerno et al., 2024) and distributed under the
CCBY4.0 license.

## Competing interests

The authors declare that they have no conflict of interest.

## Special issue statement



This article is part of the special issue "Hydrometeorological data from mountain and
alpine research catchments". It is not associated with a conference.

**Author contribution**

F.S., N.G. and N.C. drafted the article, G. T. contributed to improving the manuscript,
M.T.M. and F. D. built the Geonetwork platform, N.G. assured the data quality assess-
ment, G. V. and K. B. are the responsible for the management of the weather stations.

**Acknowledgements**

The Pyramid Meteorological Network was supported by the MIUR (Ministero
dell'Istruzione e del Merito) through Ev-K2-CNR/SHARE and CNR-DTA/NEXTDATA
project within the framework of the Ev-K2-CNR and Nepal Academy of Science and
Technology (NAST).

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



*Table 1. List of surface stations belonging to Pyramid Metereologicl Network*

| Station ID | Location | Latitude °N | Longitude °E | Elevation m a.s.l. | Mean feature of the landscape | Sampling rate |
|---|---|---|---|---|---|---|
| **AWSSC** | South Col | 27.98 | 86.76 | 7 986 | Mountain peak (off glacier) | 1 hour |
| **CNG_SNP** | Changri Nup | 27.96 | 86.93 | 5 700 | Glacier (on glacier) | 1 hour |
| **AWS4** | Kala Patthar | 27.99 | 86.83 | 5 600 | Mean glaciers surface (on glacier) | 1 hour |
| **AWS0, AWS1** | Pyramid | 27.96 | 86.81 | 5 035 | Mean glacier fronts (off glacier) | 1 hour |
| **AWS2** | Pheriche | 27.90 | 86.82 | 4 260 | Treeline (off glacier) | 1 hour |
| **AWS5** | Namche | 27.80 | 86.71 | 3 570 | Forests (off glacier) | 1 hour |
| **AWS3** | Lukla | 27.70 | 86.72 | 2 660 | Forests (off glacier) | 1 hour |


*Table 2. % of daily missing data for each variable. AT: 2m Atmospheric Temperature*
*(°C); RR: Rainfall Rate (mm); RH: Relative Humidity (%); AP: Atmospheric Pressure*
*(hPa); WS: Wind Speed (m/s); WD: Wind Direction (°)*

| Missing rate (1994/2023) (%) | AP | AT | RH | RR | WD | WS | UVA |
|---|---|---|---|---|---|---|---|
| Z7986 | 54 | 61.8 | 78 | - | 67.9 | 64.9 | 46,6 |
| Z5700 | - | 6 | 6 | - | 25.6 | 25.1 | - |
| Z5600 | 16.5 | 18.1 | 18.9 | 44.6 | 26.9 | 28.2 | - |
| Z5035 (AWS0) | 12.6 | 18.1 | 18.5 | 23.3 | 53.4 | 12.6 | - |
| Z5035 (AWS1) | 7.2 | 6.8 | 22.3 | 9.4 | 10.5 | 9.1 | - |
| Z4260 | 13 | 15.3 | 14.4 | 14.8 | 20.2 | 23.3 | - |
| Z3570 | 39 | 41.9 | 53.1 | 42.9 | 43.7 | 42.5 | - |
| Z2660 | 49.1 | 51 | 63 | 52.1 | 54 | 49.4 | - |


*Table 3. List of sensors with measurement height, manufacturer and accuracy.*

| Parameter | Sensor | Manufacturer | Accuracy |
|---|---|---|---|
| | **AWS0 (Z5035)** | | |
| Air temperature | Precision Linear Thermistor (2m) | MTX | 0.1°C |



| | | | |
|---|---|---|---|
| Precipitation | Tipping Bucket (1.5m) | MTX | 0.2 mm |
| Relative humidity | Solid state hygrometer (2m) | MTX | 3% |
| Atmospheric pressure | Aneroid capsule (2m) | MTX | 0.5hPa |
| **AWS1(Z5035)** | | | |
| Air temperature | Thermoresistance (2m) | Lsi-Lastem | 0.1°C |
| Precipitation | Tipping Bucket (1.5m) | Lsi-Lastem | 2% |
| Relative humidity | Capacitive Plate (2m) | Lsi-Lastem | 2.5% |
| Atmospheric pressure | Slice of Silica (2m) | Lsi-Lastem | 1hPa |
| **AWS4(Z5035)** | | | |
| Air temperature | Thermoresistance (2m) | Lsi-Lastem | 0.1°C |
| Precipitation | Tipping Bucket (1.5m) | Lsi-Lastem | 1% |
| Relative humidity | Capacitive Plate (2m) | Lsi-Lastem | 1.5% |
| Atmospheric pressure | Slice of Silica (2m) | Lsi-Lastem | 1hPa |
| **AWS2(Z4260)** | | | |
| Air temperature | Thermoresistance (2m) | Lsi-Lastem /Vaisala | 0.1°C/0.3°C |
| Precipitation | Tipping Bucket (1.5m) | Lsi-Lastem | 2% |
| Relative humidity | Capacitive Plate (2m) | Lsi-Lastem /Vaisala | 1.5%/2.5% |
| Atmospheric pressure | Slice of Silica (2m) | Lsi-Lastem /Vaisala | 1hPa/0.5 hPa |
| **AWS5(Z3570)** | | | |
| Air temperature | Thermoresistance (2m) | Lsi-Lastem | 0.1°C |
| Precipitation | Tipping Bucket (1.5m) | Lsi-Lastem | 2% |
| Relative humidity | Capacitive Plate (2m) | Lsi-Lastem | 2.50% |
| Atmospheric pressure | Slice of Silica (2m) | Lsi-Lastem | 1hPa |
| **AWS3(Z2660)** | | | |
| Air temperature | Thermoresistance (2m) | Lsi-Lastem /Vaisala | 0.1°C/0.3°C |
| Precipitation | Tipping Bucket (1.5m) | Lsi-Lastem | 2% |
| Relative humidity | Capacitive Plate (2m) | Lsi-Lastem /Vaisala | 1.5%/2.5% |
| Atmospheric pressure | Slice of Silica (2m) | Lsi-Lastem /Vaisala | 1hPa/0.5 hPa |
| **CNG_SNP(Z5700)** | | | |
| Air temperature | Thermoresistance (2m) | Lsi-Lastem /Vaisala | 0.1°C/0.3°C |
| Precipitation | Tipping Bucket (1.5m) | Lsi-Lastem | 2% |
| Relative humidity | Capacitive Plate (2m) | Lsi-Lastem /Vaisala | 1.5%/2.5% |
| Atmospheric pressure | Slice of Silica (2m) | Lsi-Lastem /Vaisala | 1hPa/0.5 hPa |
| **AWSSC(Z7986)** | | | |
| Air temperature | Thermoresistance (2m) | Lsi-Lastem /Vaisala | 0.1°C/0.3°C |
| Precipitation | Tipping Bucket (1.5m) | Lsi-Lastem | 2% |
| Relative humidity | Capacitive Plate (2m) | Lsi-Lastem /Vaisala | 1.5%/2.5% |
| Atmospheric pressure | Slice of Silica (2m) | Lsi-Lastem /Vaisala | 1hPa/0.5 hPa |


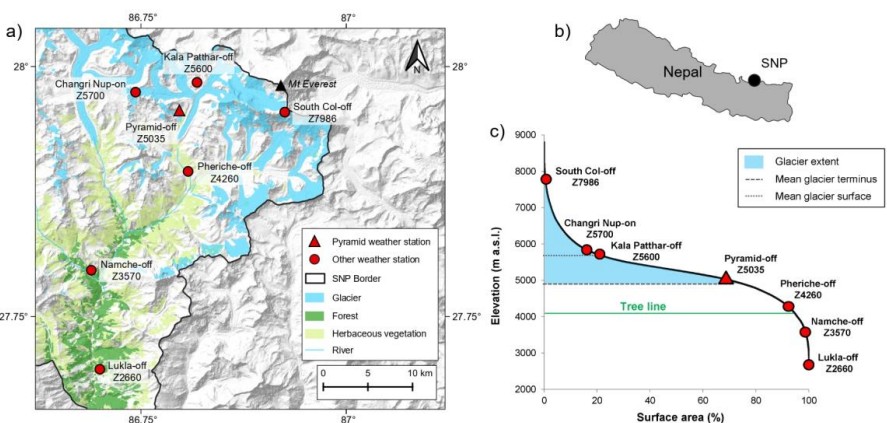


*Figure 1. a, b) Location of meteorological monitoring network in the Sagarmatha*
*National Park (SNP), Nepal c) Hypsometric curve of SNP and altitudinal glacier*
*distribution. Along this curve, the locations of meteorological stations belonging to*
*Pyramid Observatory Laboratory are presented.*







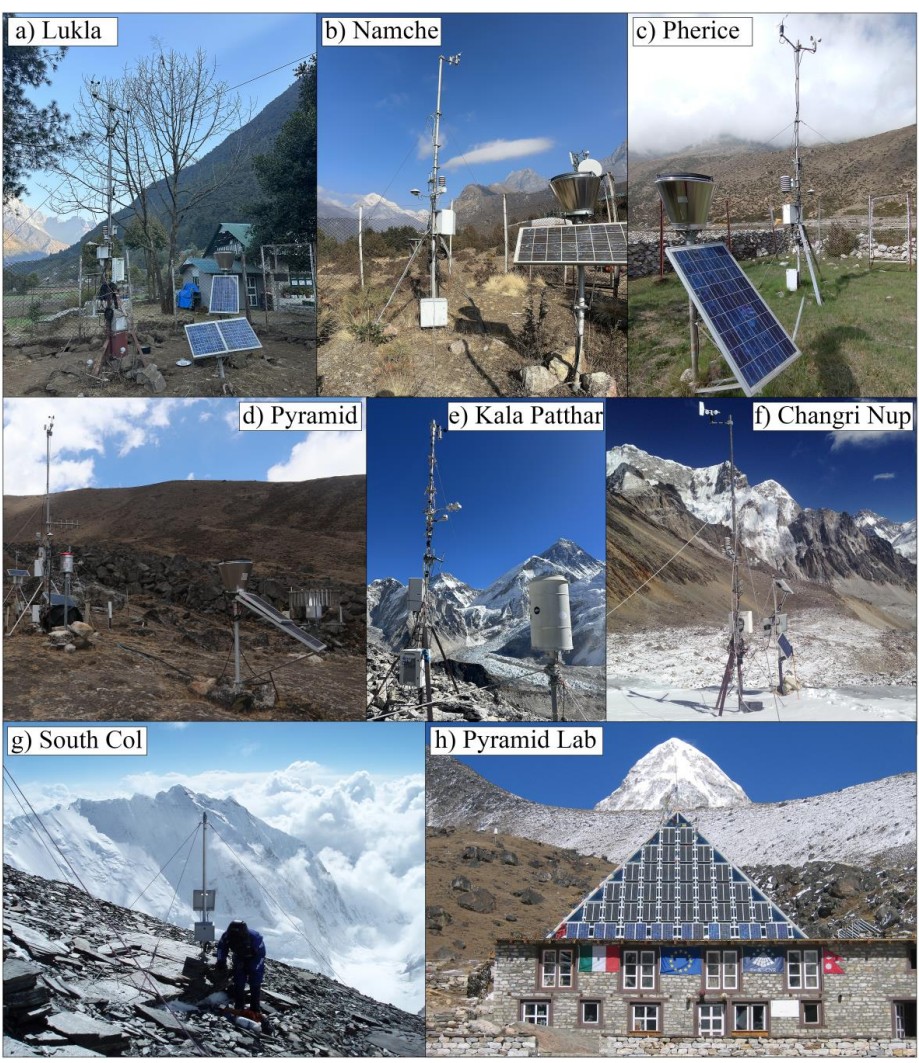


*Figure 2. Photographs of the Pyramid Meteorological Network*









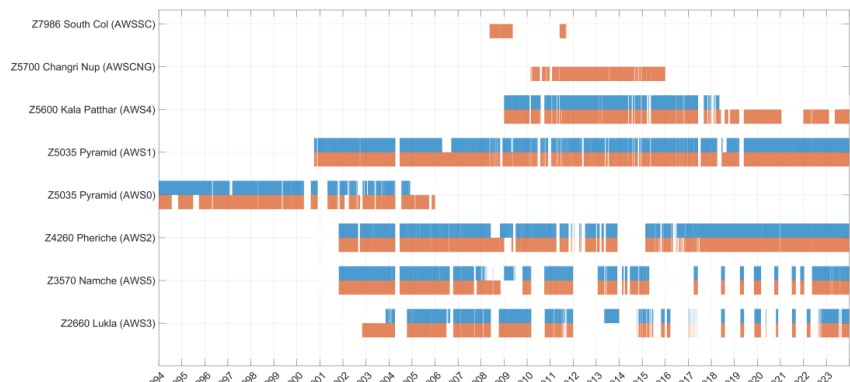


*Figure 3. Available data time series (precipitation: blue; temperature: orange) for the Pyramid Meteorological Network since 1994*

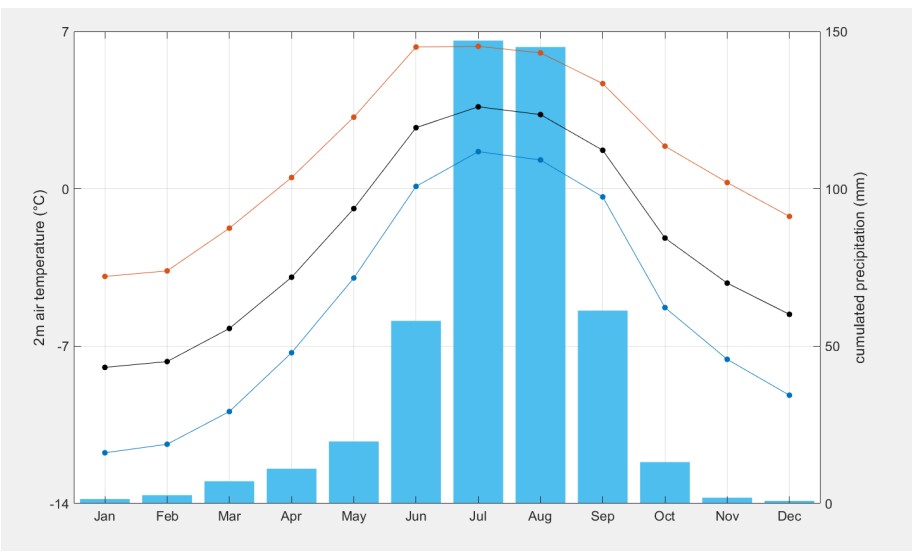


*Figure 4. Mean monthly cumulated precipitation and minimum, maximum, and mean temperature at Pyramid station (Z5035 m a.s.l. (reference period 1994–2023).*

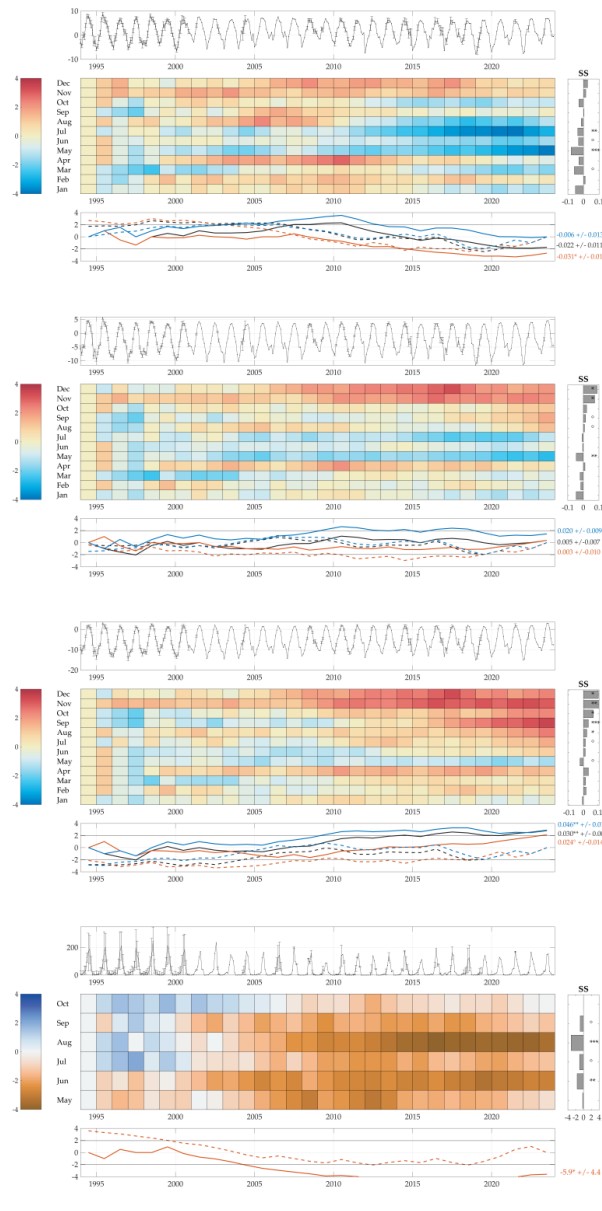


*Figure 5. Air temperature and precipitation trend analysis at Pyramid station (Z5035).*
*Complete time series for a) maximum, b) mean, c) minimum, and d) total precipitation.*
*The top graph of each meteorological variable (from a to d) shows the monthly trend. The*
*grids display the results of the MK test applied at the monthly scale and calculated from*
*the beginning of the series to the given year. The colour bar represents the normalized*
*Kendall's tau coefficient μ(τ). The colour tones below −1.96 and above 1.96 are*



*significant (α = 5%). On the right, the monthly Sen's Slope (SS) and the significance levels*
*for 1994–2023 (ˆP < 0.1, \*P < 0.05, \*\* P < 0.01, \*\*\* P < 0.001).* **The bottom graph plots**
*the progressive μ(τ) (solid lines) and retrograde (dotted line) of the seqMK test (that is,*
*calculated from the beginning or from the end, respectively, of the series to the given year)*
*for the cold season (NDJFMA) (blue), the warm season (MJJASO) (orange) and for the*
*entire year (black). For each year, below-zero lines indicate negative trends (calculated*
*from 1994).*