# Peer review of "What is climate change doing in Himalaya? Thirty"

_Earth System Science Data, 2024_

## Referee Comment (RC1)

This manuscript presents a dataset of seven meteorological stations in the Khumbu Valley, Nepalese Himalaya. The oldest of these stations dates back to 1994, and as such it is some of the very little data available at high-elevation in the Himalaya, and contains invaluable long timeseries of climate data. I commend the authors on their efforts to archive this data in way that is available to the scientific community. It will be a resource of considerable scientific importance.

As this manuscript is presenting a new geoportal to share this data, some of my comments are in relation to the portal itself, as I believe there are a few technical changes to make before it is ready for publication.

**Comments on the data and data portal**

On the data portal, it is possible to output the data as a table or a graph, but not to download the csv links from here. The zenodo links in the manuscript do contain the data, so perhaps these could be linked to the data portal? In addition, I think there is a problem with the metadata, as the html link reports an error and is not easily readable. On my screen, the metadata is also written in white text on a white background.

As the data being presented in this manuscript is the same as that used in Salerno et al., 2023, I have a question from that manuscript relating to this one, namely in Fig. 3 of Salerno et al., 2023, the authors refer to northerly flow as 90°–270° (and later to this degree range as 'southward'). This seems non-standard, so could the authors confirm that the data from the anemometers described in this manuscript are oriented in the standardised setup, with 0° pointing north and representing northerly wind (southerly flow being represented by values 90°–270°)? If not, please list a clear clarification in both this manuscript and with the wind direction data.

Could the authors please add a timezone to the data (this may be in the metadata, but it is not currently available).

**Comments on this manuscript**

Line 109-111: The authors refer to strong diurnal katabatic winds, but the windrose shown in Salerno et al., 2023 Fig. 3 suggests predominantly anabatic winds (most strong winds are South Easterly i.e. up-glacier). Is there some evidence from the data in this manuscript of the strong katabatic winds above 4500 m?

Looking at the precipitation timeseries in figure 5, the decrease in precipitation appears to be a step change around 2001, when the new AWS was installed. Could the authors comment on whether they consider this trend reliable, or potentially due to the new instrumentation? In the interpretation of precipitation trends on line 250, what is the reference period for the 41% reduction?

Figure 5: This figure is a little hard to interpret due to the many different plots and the very small size. Could the authors please enlarge the figure and label the figure (a, b, c etc, or

perhaps a) i), a) ii) etc) so that it is easier to refer to the different plots in the figure caption. The figure caption should refer to temperature for the initial reference to "a) maximum, b) mean, c) minimum". Including colour bar labels and axis labels referring to the variable being shown would also help interpretation. Please also change 'the top graph….shows the monthly trend' to 'the top graph….shows the monthly timeseries'.

Line 223: Should this trend be -0.**0**31 ± 0.015 °C y-1? It seems out by an order of magnitude compared with figure 5.

Line 229-230: the authors mention a decreasing trend in the cold months, but in line 224-227 they discuss increases/no trend in the cold months. Please clarify this.

Line 238: For the minimum air temperature, fig. 5 shows a positive (rather than negative) trend, statistically significant with p<0.01.

Figure 1: The location of Changri Nup station does not match the location in the table 1. Could the authors please check all the station locations in figure 1.

Line 262: please clarify in the conclusions that this decreasing trend in air temperature only holds for maximum air temperature in certain months.

**Minor comments:**

103 and figure 4 caption: typo cumulated precipitation -> cumulative precipitation

Table 3: Please standardise the names of the stations e.g. include the AWS4 etc IDs in table 3. It would also be helpful to explicitly state that the Z naming convention relates to elevation of the AWS, as it's not immediately obvious.

Line 212: I think it would be more applicable to reference the minimum temperature being mostly above 0 here, as in principle the mean temperature could be above 0 but precipitation fall as snow during the night.

247:  typo period -> periods

---

## Referee Comment (RC2)

**Review of "What is climate change doing in Himalaya? Thirty years of the Pyramid Meteorological Network (Nepal)" written by Franco Salerno and 8 others.**

The author published the one of the very interesting and important AWS data set from different elevation AWSs from the Khumbu region, well researched are in the Himalaya. The work done on Khumbu since 1990s, stablishing the network of stations and maintaining them for long term is one of the challenging and expensive work. These data have been used in many work and insightful results are already published and would be very helpful for the further research in the region. I would like to thanks to the all the hands who have been involved in the work since the beginning to till date for the development, maintaining and collecting data in these works.

Comments:

As there are overlapping 2000-2007 of the AWS data at 5035 elevation, and at that period the temperature data are not similar to each other. Do you have any idea why they are not similar?

The maximum temperature trend in both here and Salerno et al 2023 is decreasing after 2000, when there is a new station installed at pyramid. This might be because of the new stations. What do you define this?

Also different international groups has installed the AWS at Pyramid since many years, have tried to compare the data from EvK2CNR networks data to other stations from the same place?

There was big/typhoon events in October 2013 and 2014 (Shea et al 2015), around 80 and 40 mm of precipitation. In the data from pyramid I think its not visible. I believe these kind of big events and precipitation would play important role for climate analysis and modeling. How do other user would incorporate such problem in data? Or how the public can identify such error in future? Any suggestion or comments page for the public users?

The data is easy to visualize in https://geoportal.mountaingenius.org/portal/ but not downloadable form here.  So do you have any plans to modify it or you also keep the zenodo links for the download.

Line by line comments:

L41: the reverence should be updated with the recent one

L57: correct the reference

L90: add some old reference from the Japanese researcher

L101-103: the 90% of precipitation during the monsoon is quite different than the result from Khadka et al 2022 and shea et al 2015, quite surprising.

L131: new one installed during 2022 at 8810 m?

L134: I think its better to modify the reverence as the Wagnon et al 2021 focused only on Mera Glacier.

L441: is that the kala pathhar aws is in the glacier

Table 2: it would be easier to follow I the order of the awss is same as table 1 and same for the table 3.

Table 3: is the aws4(z5035) is at 5035m?

---

## Author Response (AR1)

Comment to the Editor
We thank the Editor for the thorough assessment of the manuscript. We replied to each individual area of concern expressed by the Reviewers as it can be read below.

Response to Reviewer #3
Reviewer comments shown as "RC:", author replies as "AR:".

RC: Review of Salerno and Guyennon et al. for ESSD data description paper
General comments
The manuscript describes and make available an important dataset of weather station data at the elevation of glacier ablation zones in the region. The dataset is useful, unique and I highly recommend that the data is made freely available. However, I don't think the manuscript is sufficient to support the publication of the dataset in the current version.
AR: We thank the Reviewer for the thorough assessment of the manuscript. All comments have been addressed, providing a point-to-point response to each comment.

RC: Specific comments
The manuscript is structured like a regular research article – this is not necessary for the ESSD dataset description format. I think this might be a contributing factor to why I find the manuscript hard to use as a dataset description paper in the current state. Overall, I think the paper could be restructured to a structure like this:
Introduction
Area description
Data description
General AWS description
Specific details on maintenance and precipitation undercatch etc..
If the data is published with filled gaps, then the gap filling should go here.
Statistical analysis
Statistical method
If gap filling is done only for the purpose of the statistical method then the description should go here
Results from the statistical analysis
Discussion of the statistical analysis
Conclusion of the statistical analysis
Data availability
Overall conclusion (mainly just summing up)
I don't think the data portal description belongs in an ESSD dataset description paper, but I will leave this up to the editor to decide.
AR: The overall text has been adapted considering the proposed structure by the reviewer. We clarified that we are publishing two kinds of data: filled monthly time series for Tmax,Tmin, Tmean, and Prec (1994-2023) and hourly raw data for all variables (Air temperature, Precipitation, Relative humidity, Atmospheric pressure).We inserted the following new sections: "3.2 Data description" section and "5 Discussion of the statistical analysis" section

RC: I hope the following comments will help clarify:
Section 3.1 gives a nice overview of the location and history of the AWSs and instruments incl. uncertainty is listed in Table 3. However, here I am missing more detail on the instrument's maintenance schedule, recalibration of hygrometer and a short description of under catch in the precipitation gauge (this is mentioned later but it would be nice to have it already here).
AR: The suggestion has been followed and these parts have been discussed in the new 3.2 section.

RC: Section 3.3 heavily relies on the description of gap filling method in Salerno et al., 2015 which makes the section hard to understand when you have not read the paper. So, I think the method for gap filling should be elaborated here so this section can stand by itself.

AR: In Salerno et al., 2025 we decided to put the details on the filling method in the Supplementary Material because it is articulated, although it is well described in all its details. We think that if the reader is interested in the adopted methodology he can easily access this material. Furthermore, we think that copy and paste this method in the main text of this paper could compromise the readability. On other option could be to create a new Supplementary Section for this paper, but we leave the Editor to decide on this.

RC: Figure 5: In the total precipitation there is clearly a shift in monthly trend data around 2001. I guess this is likely to be due to a change in instrumentation or logging frequency or something else – but this should be discussed in the main text.

AR:We improved the readability of figure 5, so that the errorbar associated with the uncertainty of the reconstruction does not interfere with the readability of the month mean values (see also figure R3). Over the 1994-2001 period, the reconstruction relied almost totally on the AWS0 station. During their common time of observation, AWS0 and AWS1 measured highly similar data (daily total precipitation correlation between the two stations is 0.97, n=2298), but with some differences. In the reconstruction (fig. 5), the systematic bias in the daily precipitation distribution between AWS0 and AWS1 has been corrected with a quantile mapping regression (e.g. Déqué 2007). The uncertainty associated with the quantile mapping is monitored, together with uncertainty of the multiple imputation for further missing data. Details are given in Salerno et al. (2015, supplementary material (https://tc.copernicus.org/articles/9/1229/2015/tc-9-1229-2015-supplement.pdf). We think that the step change impression in 2001 is more a consequence of the visual impact of the uncertainty associated with AWS0 imputation. We are confident in the trend estimation reliability, with a known large uncertainty on the associated intensity (-5.9 +/- 4.4). It is worth noting that such a result is in line with the impact of increased katabatic winds on the precipitation reduction due to a downward shift of the convergence elevation as described in Lin et al. (2021) and Salerno et al. (2023).

The manuscript has been corrected adding the uncertainty associated with the precipitation trend.

[Figure]

Figure R3 Pyramid monthly precipitation and associated gap filling uncertainty. The visual impact of uncertainty has been reduced to avoid confusion.

RC: Line by line

38-41: This is interesting – but I don't think it is relevant in the context of this manuscript.

AR: We think that monitoring how climate changes at high elevation is useful also for human systems and Alpine biomes. Glaciers are only one of the elements impacted by climate change..

RC: 50: When you write "few" maybe you mean "only few"?

AR: Modified as suggested.

RC: 52: It is mentioned section 3.1 line 128-134 that there are stations at higher elevation than the stations mentioned here. Maybe this statement belongs better here in the introduction?

AR: Modified as suggested.

RC: 167-168: Perhaps reformulate: Pyramid station has data gaps corresponding to ca. 10 and 15 % of …

AR: Modified as suggested.

RC: 262-264: I believe this statement belongs in the results and discussion section especially when referring to another study. One possibility is to split results and discussion in two subsections to make it possible to summarize the findings in the results in the discussion and compare to what other have found.

AR: The suggestion has been accepted and a discussion section has been created.

RC: Figures and tables

Figure 5: The panels should be marked with a) b) c) and d) and units should be on the y-axis of all subpanels.

AR: We edited Fig. 5 and its caption to improve readability.

RC: Table 2: The caption should be changed so that it starts with the word Percentage (and not %). The first field in the table should describe what is in the top row and first column currently it is just a mini-caption.

AR: Done.

Response to Reviewer #2
Reviewer comments shown as "RC:", author replies as "AR:".

RC: The author published the one of the very interesting and important AWS data set from different elevation AWSs from the Khumbu region, well researched are in the Himalaya. The work done on Khumbu since 1990s, stablishing the network of stations and maintaining them for long term is one of the challenging and expensive work. These data have been used in many work and insightful results are already published and would be very helpful for the further research in the region. I would like to thanks to the all the hands who have been involved in the work since the beginning to till date for the development, maintaining and collecting data in these works.
AR: We thank the Reviewer for the valuable suggestions to improve and strengthen the paper. All comments have been thoroughly addressed in our responses below.

RC: As there are overlapping 2000-2007 of the AWS data at 5035 elevation, and at that period the temperature data are not similar to each other. Do you have any idea why they are not similar?
AR: The temperature data measured by the two AWSs at 5035m (i.e. AWS0 and AWS1) during their overlapping period appears highly similar (r2=0.93, mean bias =-0.073 °C, figure R4 upper and right panels) but show some scatter mostly during the winter from november to march, despite being the season presenting the lowest bias (figure R4 lower left panel). The reasons for those differences could reasonably come from the lower sampling rate at AWS0 (2 hours). At daily scale, the scatter almost disappears (Fig. R4 upper right panel).

[Figure]

Figure R4 comparison between hourly AT at AWS0 and AWS1

RC: The maximum temperature trend in both here and Salerno et al 2023 is decreasing after 2000, when there is a new station installed at pyramid. This might be because of the new stations. What do you define this?
AR: As reported in figure R4, the two stations AWS0 and AWS1 did not present mean biases in their temperature observations that could explain the decreasing trend (oppositely, AWS0 has been measuring slightly lower temperature than AWS1 during the years of common operation). In any case, those biases have been corrected by a quantile mapping applied at daily scale (as described in details in Salerno et al., 2015, https://tc.copernicus.org/articles/9/1229/2015/tc-9-1229-2015-supplement.pdf) for the reconstructions used in Salerno et al., 2023 and presented here. Moreover, as it can be quantified by the sequential mann kendall (both in fig.1 of Salerno et al 2023 and figure 5 of the current manuscript), the changing point occurs later between 2007 and 2015. The trend uncertainty associated with the reconstruction process has been estimated (method in Salerno et al., 2015) and

reported. The authors are confident in the fact that the summer maximum temperature trend is not a result of the new station AWS1. It is worth noting that similar decreasing Tmax summer trends have also been observed by independent dataset in other Himalayan high elevation AWSs close to glacier masses (Salerno et al., 2023, Fig. S5). Moreover, our findings are corroborated by the trend observed for the last 20 years even at Pheriche station (Fig. S4 in Salerno et al., 2023).

RC: Also different international groups has installed the AWS at Pyramid since many years, have tried to compare the data from EvK2CNR networks data to other stations from the same place?
AR: Lines 132-138 report the other meteorological networks located in the same valley. However, there is not an overlap in the same place, so the comparison has not been performed.

RC: There was big/typhoon events in October 2013 and 2014 (Shea et al 2015), around 80 and 40 mm of precipitation. In the data from pyramid I think its not visible. I believe these kind of big events and precipitation would play important role for climate analysis and modeling. How do other user would incorporate such problem in data? Or how the public can identify such error in future? Any suggestion or comments page for the public users?
AR: We are aware of the October 2023 extreme event (some of the authors were actually there during the event). Unfortunately, the precipitation gauge is not heated, and during those events the precipitation was solid. Nevertheless, looking at lvl1 AWS1 hourly data (figure R5), the event of October 2013 can be seen, and it was also captured by snow level observations, which are still under validation and will hopefully be released in the coming years.

[Figure]

Figure R5 Pyramid hourly precipitation and snow level (not shown in this paper nor currently present in the portal) during 2013 and 2014.

RC: The data is easy to visualize in https://geoportal.mountaingenius.org/portal/ but not downloadable form here. So do you have any plans to modify it or you also keep the zenodo links for the download.
AR: Thank you for pointing out these issues, which have been solved. In the updated version of the Geoportal, the download of the csv file is permitted under registration. The link to Zenodo allows access to the section of the portal with these data. Moreover, the link to the metadata file has been changed (https://zenodo.org/records/15211352).

RC: Line by line comments: L41: the reverence should be updated with the recent one

AR: We added The GlaMBIE Team, 2025.

RC: L57: correct the reference
AR: Corrected.

RC: L90: add some old reference from the Japanese researcher
AR: Done.

RC: L101-103: the 90% of precipitation during the monsoon is quite different than the result from Khadka et al 2022 and shea et al 2015, quite surprising.
AR: From the mean climatology at Pyramid (figure 4) , 87.7% of precipitation falls from June to September. The reference "*around 90%*" has been replaced by the actual value (87.7%) in the text L217.

[Figure]

Figure R6 Mean annual cumulated precipitation from pyramid climatology: the cumulated precipitation rises from 8.94% to 96.61% of the total annual between June and September.

RC: L131: new one installed during 2022 at 8810 m?
AR: done

RC: L134: I think its better to modify the reverence as the Wagnon et al 2021 focused only on Mera Glacier.
AR: Done. We added "*e.g., *"

RC: L441: is that the kala pathhar aws is in the glacier
AR: It was a typo. The error has been corrected.

RC: Table 2: it would be easier to follow if the order of the Awss is same as table 1 and same for the table 3.
AR: We followed the suggestion and the order of Aws in Table 3 has been corrected

RC: Table 3: is the aws4(z5035) is at 5035m?
AR: The error has been corrected

Response to Reviewer #1
Reviewer comments shown as "RC:", author replies as "AR:".

RC: This manuscript presents a dataset of seven meteorological stations in the Khumbu Valley, Nepalese Himalaya. The oldest of these stations dates back to 1994, and as such it is some of the very little data available at high-elevation in the Himalaya, and contains invaluable long timeseries of climate data. I commend the authors on their efforts to archive this data in way that is available to the scientific community. It will be a resource of considerable scientific importance.
As this manuscript is presenting a new geoportal to share this data, some of my comments are in relation to the portal itself, as I believe there are a few technical changes to make before it is ready for publication.
AR: We thank the Reviewer for the constructive feedback and the thorough assessment of the manuscript. All comments have been addressed, providing a point-to-point response to each comment.

RC: Comments on the data and data portal
On the data portal, it is possible to output the data as a table or a graph, but not to download the csv links from here. The Zenodo links in the manuscript do contain the data, so perhaps these could be linked to the data portal? In addition, I think there is a problem with the metadata, as the html link reports an error and is not easily readable. On my screen, the metadata is also written in white text on a white background.
AR: Thank you for pointing out these issues, which have been solved. In the updated version of the Geoportal, the download of the csv file is permitted under registration. The link to Zenodo allows access to the section of the portal with these data. Moreover, the link to the metadata file has been changed (https://zenodo.org/records/15211352).

RC: As the data being presented in this manuscript is the same as that used in Salerno et al., 2023, I have a question from that manuscript relating to this one, namely in Fig. 3 of Salerno et al., 2023, the authors refer to northerly flow as 90°–270° (and later to this degree range as 'southward'). This seems non-standard, so could the authors confirm that the data from the anemometers described in this manuscript are oriented in the standardised setup, with 0° pointing north and representing northerly wind (southerly flow being represented by values 90°–270°)? If not, please list a clear clarification in both this manuscript and with the wind direction data.
AR: Thank you for this comment, which allows us to clarify this point. We confirm that the data stored in the repository of this paper are oriented in the standardised setup, with 0° pointing to the North and representing northerly winds. The caption of Figure 3 in Salerno et al. (2023) "...*downward wind speed among hours of northerly (90°–270°) flow at Changri...*" can be misleading, since the authors wrote "*90°–270°*" in order to indicate the North/South division as hemispheres. However, we agree that reporting "*270°-90°*" would have been more clear. To avoid any confusion, the following sentence has been added in the metadata: "*Wind direction follows the standard definition, with 0° pointing North and representing northerly wind*".

RC: Could the authors please add a timezone to the data (this may be in the metadata, but it is not currently available).
AR: The data are shared in local time (i.e., Nepal Standard Time NPT), defined as UTC+5:45. The information has now been reported in the metadata: "*All times are local time (Nepal Standard Time (NPT), UTC + 5:45)*".

RC: Comments on this manuscript
Line 109-111: The authors refer to strong diurnal katabatic winds, but the windrose shown in Salerno et al., 2023 Fig. 3 suggests predominantly anabatic winds (most strong winds are South Easterly i.e.

up-glacier). Is there some evidence from the data in this manuscript of the strong katabatic winds above 4500 m?

AR: We wrote "*strong diurnal katabatic winds*" referring to the intensity of downwards winds during diurnal hours (as evidenced in Salerno et al., 2023 Fig. 3b), while in terms of frequency, most strong winds are up valley, as those analyses refer to the warm season, under the monsoon influence. Evidence of strong diurnal katabatic winds can be seen from the provided Changri Nup station data. Figure R1 (left panel) shows common downslope direction (indicatively 330°-60°) between nighttime and diurnal time, oppositely to diurnal upvalley winds (indicatively 120°-180°). The frequency of main downslope directions increases during diurnal hours for warmer days. The right panel shows that north east winds (30°-60°) present a peak in intensity during early afternoon (13:00-15:00), that we interpret as diurnal katabatic winds from the glacier north east to the Changri Nup on glacier station. Please note that fig. 3c in Salerno et al. (2023) reports the mean integral of 270°-90° of figure R1 right panel (excluding low wind speed).

[Figure]

Figure R1. Evidence of strong diurnal katabatic winds during the warm period (MJJASO months). Left panel reports the frequency of wind directions separated between day and night time. The frequency of diurnal distribution is also reported for *warmer days* (intended as ERA 300 hpa air temperature positive anomalies). Right panel reports the mean intensity (wind speed, colorcode) as a function of wind direction and hour of the day.

Evidence of diurnal katabatic winds can be further appreciated by looking at the trends at the Pyramid Station (figure R2), with increasing diurnal northern (270°-90°) winds during the past decade.

[Figure]

Figure R2 Wind speed means intensity (colorcode) as a function of wind direction and hour of the day for the period 1994-1999, 2000-2007, 2008-2014 and 2015-2023. Black circles highlight the increasing intensity of downward (90°-270°) diurnal winds during the last decade.

RC: Looking at the precipitation timeseries in Figure 5, the decrease in precipitation appears to be a step change around 2001, when the new AWS was installed. Could the authors comment on whether they consider this trend reliable, or potentially due to the new instrumentation? In the interpretation of precipitation trends on line 250, what is the reference period for the 41% reduction?
AR: We improved the readability of figure 5, so that the errorbar associated with the uncertainty of the reconstruction does not interfere with the readability of the month mean values (see also figure R3). Over the 1994-2001 period, the reconstruction relied almost totally on the AWS0 station. During their common time of observation, AWS0 and AWS1 measured highly similar data (daily total precipitation correlation between the two stations is 0.97, n=2298), but with some differences. In the reconstruction (fig. 5), the systematic bias in the daily precipitation distribution between AWS0 and AWS1 has been corrected with a quantile mapping regression (e.g. Déqué 2007). The uncertainty associated with the quantile mapping is monitored, together with uncertainty of the multiple imputation for further missing data. Details are given in Salerno et al. (2015, supplementary material (https://tc.copernicus.org/articles/9/1229/2015/tc-9-1229-2015-supplement.pdf). We think that the step change impression in 2001 is more a consequence of the visual impact of the uncertainty associated with AWS0 imputation. We are confident in the trend estimation reliability, with a known large uncertainty on the associated intensity (-5.9 mm/y +/- 4.4). It is worth noting that such a result is in line with the impact of increased katabatic winds on the precipitation reduction due to a downward shift of the convergence elevation as described in Lin et al. (2021) and Salerno et al. (2023). The reference period has been added (1994-2023) as suggested by the reviewer.
Furthermore, the manuscript has been corrected adding the uncertainty associated with the precipitation trend.

[Figure]

Figure R3 Pyramid monthly precipitation and associated gap filling uncertainty. The visual impact of uncertainty has been reduced to avoid confusion.

RC: Figure 5: This figure is a little hard to interpret due to the many different plots and the very small size. Could the authors please enlarge the figure and label the figure (a, b, c etc, or perhaps a) i), a) ii) etc) so that it is easier to refer to the different plots in the figure caption. The figure caption should refer to temperature for the initial reference to "a) maximum, b) mean, c) minimum". Including colour bar labels and axis labels referring to the variable being shown would also help interpretation. Please also change 'the top graph….shows the monthly trend' to 'the top graph….shows the monthly timeseries'.
AR: Figure 5 and its caption have been edited to improve readability.

RC: Line 223: Should this trend be -0.031 ± 0.015 °C y-1? It seems out by an order of magnitude compared with figure 5.
AR: This was a typo and it has been fixed.

RC: Line 229-230: the authors mention a decreasing trend in the cold months, but in line 224-227 they discuss increases/no trend in the cold months. Please clarify this.

AR: The suggestion has been followed and the text clarified.

RC: Line 238: For the minimum air temperature, fig. 5 shows a positive (rather than negative) trend, statistically significant with $p<0.01$.
AR: It was a typo. Amended.

RC: Figure 1: The location of Changri Nup station does not match the location in the table 1. Could the authors please check all the station locations in figure 1.
AR: The typo was corrected.

RC: Line 262: please clarify in the conclusions that this decreasing trend in air temperature only holds for maximum air temperature in certain months.
AR: The suggestion has been followed and the text clarified.

RC: Minor comments:
103 and figure 4 caption: typo cumulated precipitation -> cumulative precipitation
AR: Done.

RC: Table 3: Please standardise the names of the stations e.g. include the AWS4 etc IDs in table 3. It would also be helpful to explicitly state that the Z naming convention relates to elevation of the AWS, as it's not immediately obvious.
AR: The suggestion has been followed.

RC: Line 212: I think it would be more applicable to reference the minimum temperature being mostly above 0 here, as in principle the mean temperature could be above 0 but precipitation fall as snow during the night.
AR: Done.

RC: 247: typo period -> periods
AR: Done.

---

## Referee Report (RR1)

Thank you to the authors for their detailed reviewer responses.

Given that all reviewers mentioned the potential effects of station change between AWS0 and AWS1, I would recommend including a plot for precipitation, Tmax, Tmean and Tmin showing the mean monthly bias in the overlapping period (corresponding to the data shown in figure 5) in the paper. Something similar to the bottom left plot of figure R4 would be helpful. While the correlation between the two stations is important, it will not necessarily affect the trends, so it is more important to show the bias. The revised figure R3 still appears to have a change in precipitation roughly at the time of the newly installed AWS, but of course this could be a coincidence if there is little bias in the overlapping period.

For the final version could the authors ensure that figure 5 is very high resolution, as some numbers are currently difficult to read, even when zoomed in.

Please ensure that any corrections to station location are also included in the geoportal metadata and readme file from the zenodo link, which I think still needs updating.

I'd recommend the authors change 'diurnal' to 'daytime', where that is what's meant, as otherwise it may be confused with diurnal cycles of the variables (covering both day and night).

---

## Referee Report (RR2)

I would like to thank the Authors for taking all my and other reviewers' comments into consideration – I find the updated version of the paper improved, and I only have few minor comments.

Line by line:

71: "Here, we present the database in which all meteorological data are stored..."

A data paper in ESSD should present data – not a database – the database is just described here because it is a useful tool to get the data: Please reformulate so that it is clear that you primarily present a dataset, but of course include the info already stated that you are also presenting the database as well as explore small-scale climate variability.

124-127: Explain briefly the Z in: (Z2660), (Z3570) ...?

138: Again, I would call it the dataset in stead of the database as to me database refers to a piece of software.

---

## Author Response (AR2)

Dear Editors,

thanks a lot for your suggestions.

All your comments have been addressed and included in the new version of the manuscript. In particular, we added Supplementary Information.

Furthermore, we provided to include the few suggestions received from the reviewer.

Best Regards

Franco Salerno and Nicolas Guyennon

---

## Author Response (AR3)

Dear Editors,

Thanks for your suggestions.

We provided to integrate the last comment.

Best Regards

Franco Salerno and Nicolas Guyennon